# The Extract of *Scutellaria baicalensis* Attenuates the Pattern Recognition Receptor Pathway Activation Induced by Influenza A Virus in Macrophages

**DOI:** 10.3390/v15071524

**Published:** 2023-07-08

**Authors:** Mingrui Yang, Luyao Ma, Rina Su, Rui Guo, Na Zhou, Menghua Liu, Jun Wu, Yi Wang, Yu Hao

**Affiliations:** 1School of Life Sciences, Beijing University of Chinese Medicine, Beijing 102488, China; mingruiyang@126.com (M.Y.); 18616104187@163.com (L.M.); surina1437@163.com (R.S.); ruiguo9@163.com (R.G.); zhouna969222@163.com (N.Z.); liumenghuapro@163.com (M.L.); wujunccg1973@163.com (J.W.); 2Experimental Research Center, China Academy of Chinese Medical Sciences, Beijing 100700, China

**Keywords:** influenza A virus, inflammatory response, transcriptome analysis, extract of *Scutellaria baicalensis*, pattern recognition receptor

## Abstract

The dual strategy of inhibiting the viral life cycle and reducing the host inflammatory response should be considered in the development of therapeutic drugs for influenza A virus (IAV). In this study, an extract of *Scutellaria baicalinase* (SBE) containing seven flavonoids was identified to exert both antiviral and anti-inflammatory effects in macrophages infected with IAV. We performed transcriptome analysis using high-throughput RNA sequencing and identified 315 genes whose transcription levels were increased after IAV infection but were able to be decreased after SBE intervention. Combined with Gene Ontology (GO) and Kyoto encyclopedia of genes and genomes (KEGG) enrichment analysis, these genes were mainly involved in TLR3/7/8, RIG-I/MDA5, NLRP3 and cGAS pattern recognition receptor (PRR)-mediated signaling pathways. SBE inhibited the transcription of essential genes in the above pathways and nuclear translocation of NF-κB p65 as confirmed by RT-qPCR and immunofluorescence, respectively, indicating that SBE reversed PR8-induced over-activation of the PRR signaling pathway and inflammation in macrophages. This study provides an experimental basis for applying *Scutellaria baicalensis* and its main effects in the clinical treatment of viral pneumonia. It also provides novel targets for screening and developing novel drugs to prevent and treat IAV infectious diseases.

## 1. Introduction

Influenza viruses are enveloped, segmented, negative-sense, single-stranded RNA viruses known to cause annually recurrent respiratory disease in humans with a significant burden on human health and the economy [1]. In particular, the circulation of the influenza A virus (IAV) is known to generate new strains or subspecies through antigenic drift and antigenic shift, resulting in recurring seasonal epidemics and even pandemics [2,3]. IAV infection not only causes moderate respiratory illness but also leads to lower respiratory tract disease or pneumonia and even progresses to acute respiratory distress syndrome (ARDS) and leading to death from respiratory failure [4,5]. The existing intervention strategies for IAV infection, mainly vaccination and antiviral therapy, have exerted erratic efficacy due to the occurrence of antigenic variation and resistant variants [6,7]. Moreover, besides intrinsic damage to host cells caused by uncontrolled viral replication and transmission, the overly exaggerated immune response can worsen the severity of lung injury and play a more important role in pathogenesis [8]. Therefore, novel drugs or approaches for treating lung injury induced by IAV are needed to further develop based on targeting an unbalanced immune response in addition to antivirals.

*Scutellaria baicalinase* Georgi, commonly named HuangQin in Chinese, is a plant of the genus *Lamiaceae*, and its root is the main part used in medicine [9,10]. Since the chemical composition and pharmacological effects have been elucidated, *Scutellaria baicalensis* has extensive development prospects in the treatment of a wide range of diseases with pharmacological activities, including anti-inflammatory, antibacterial, antiviral, antioxidant, and anti-tumor properties [11,12,13]. Previous reports illustrated that *Scutellaria baicalensis* reduced virus titer, viral nucleoprotein (NP), proinflammatory cytokines tumor necrosis factor (TNF-α), interleukin (IL)-1, IL-6 and increased the expression of anti-inflammatory cytokines IL-10, interferon (IFN)-γ, thereby alleviating the lung injury caused by IAV infection [14,15,16,17]. Among the main active ingredients in *Scutellaria baicalensis*, the pharmacological effects of baicalin have been widely studied, demonstrating their inhibition effect on IAV replication or inflammation response via interfering with neuraminidase and host IFN-γ production [18,19]. The mentioned studies have partially explained the pharmacological effects of *Scutellaria baicalensis* or its active components, but the multi-target effects of *Scutellaria baicalensis* in the replicative pathological process of lung infection caused by IAV have not yet been clearly explained. In particular, the mechanism behind the *Scutellaria baicalensis* inhibiting effect on excessive inflammatory response remains unclear.

In the process of IAV infection, lung resident and monocyte-derived macrophages, which phagocytize virus particles and infected cells and secrete various cytokines, are essential to the innate immune response and excessive inflammation [20,21]. Regulating the macrophages effectively decreases lung damage induced by IAV, suggesting that macrophages are the suitable target and model for studying drugs to treat viral inflammation [22]. In this paper, we detected the intervention effect of an extract from *Scutellaria baicalensis* (SBE) on viral titer and inflammatory response of mice J774A.1 macrophage induced by IAV strain. We systematically analyzed the genome-wide transcriptomic profiles of cellular mRNA using the RNA-sequencing approach. The data identified that SBE reversed widespread alteration in cellular mRNA expression caused by IAV infection and highlighted essential genes in pattern recognition pathways that may act as targets of SBE, mediate the inhibition of excessive release of pro-inflammatory factors and alleviate the inflammatory response. This study may shed light on the inflammatory biology of IAV infection, reveal the mechanism of *Scutellaria Scutellaria* as a herb in treating viral pneumonia, and provide a new perspective for developing antiviral therapies against IAV.

## 2. Materials and Methods

### 2.1. Cells, Virus and SBE Solution

The mouse monocyte macrophage (J774A.1 cell line) was purchased from Procell Life Science and Technology Co., Ltd. (Wuhan, China) and cultured into high-glucose DMEM medium (No.11965118, ThermoFisher Scientific, Waltham, MA, USA) containing 10% fetal bovine serum (No.35-079-CV, Corning, Somerville, MA, USA) and 1% antibiotic penicillin-streptomycin (No. YM-S1033, Corning, USA) at 37 °C under 5% CO_2_. The mouse-adapted IAV PR8 strain (strain A/Puerto Rico/8/1934 (H1N1)) was propagated in 9-day-old chicken embryos at 35 °C for 48 h, then at 4 °C for 12 h and stored in the refrigerator at −80 °C in our laboratory. The titer of the virus was 10^−5.4^/0.1 mL in J774A.1 cell using 50% tissue culture infectious dose (TCID_50_) as standard. The SBE solution is provided by Shanghai Kaibao Pharmaceutical Co., Ltd. (No.1808320, Shanghai, China) and obtained by methanol extraction under ultrasonication from dried *Scutellaria baicalensis* roots. Before experiments, the SBE solution was filtered by 0.22 µm micro membranes and the major components were identified using ultra-performance liquid chromatography-quadrupole time-of-flight mass spectrometry (UPLC/Q-TOF MS).

### 2.2. Samples Preparation and RNA Extraction

J774A.1 cells were seeded in six-well plates (1 × 10^6^ cells/well) and divided into Mock, PR8 and SBE groups. The cells in the Mock group were not infected. Cells in the PR8 group were infected with 100 TCID_50_ PR8 strain in serum-free DMEM medium for 1 h. Cells in the group SBE were infected by the PR8 virus and then intervened with the maximum nontoxic concentration (TC_0_) at 6.25 μg/mL of SBE. The TC_0_ was obtained by the Cell Counting Kit-8 assay (CCK8) to detect the inhibition rate of SBE on J774A.1 cell, and the calculation standard was to ensure that the cell survival rate was more than 90%. After washing with phosphate-buffered saline (PBS) three times, the cells were cultured at 37 °C with a complete medium for 24 h. Then the cells were lysed with TRIzol reagent (No.15596, Thermo Fisher, USA) for RNA extraction. Three biological replicates were set for each group.

### 2.3. Generation of cDNA Library and RNA Sequencing

The RNA sequencing was performed by Qiantang Life Science and Technology Co., Ltd. (Suzhou, China). The quality control standards for RNA samples following three aspects: RNA integrity and DNA contamination were analyzed by agarose gel electrophoresis; RNA purity was determined using a nanophotometer spectrophotometer; accurate RNA integrity was detected by Agilent 2100 bioanalyzer (Santa Clara, CA, USA). The cDNA library was generated using NEBNext Ultra RNA Library Prep Kit for Illumina (San Diego, CA, USA). The first strand cDNA was synthesized in M-MuLV reverse transcriptase system using random oligonucleotide as a primer. The RNA strand was decomposed by RNase H, followed by second strand cDNA synthesis in the DNA polymerase I system. The purified double-strand cDNA was end-repaired, added with A-tail and connected with sequencing adaptors. Then AMPure XP beads were used for screening and purification to get about 250–300 bp cDNA, and finally, the library was obtained, and quality evaluated. Library sequencing was conducted on Illumina HiSeqTM2000 by following the manufacturer’s instructions.

### 2.4. Sequencing Data Quality Control and Genome Mapping

The quality control of raw reads was performed using the Fastp software to ensure the quality and reliability of data analysis. The following reads were filtered out: (1) the reads containing adaptors, (2) the reads with N content of more than 5%, and (3) low-quality reads (reads with a base quality score of less than 5 accounted for more than 50% of the total bases as low-quality reads). The clean reads were mapped to the mouse reference genome using Hierarchical Indexing for Spliced Alignment of Transcripts 2 (HISAT2) software.

### 2.5. Gene Expression Analysis

Analysis of gene expression levels was calculated using the fragments per kilobase million mapped reads (FPKM) method by StringTie and Ballgown software. Differentially expressed genes (DEGs) analysis between Mock, PR8 and SBE groups was based on the gene transcription level by DESeq2 using Negative Binomial distribution as modes for read counts. DEGs were identified based on the filtering parameters that ∣log_2_(fold change)∣ > 1 and *p*-value < 0.05.

### 2.6. Gene Ontology (GO) Term and Pathway Enrichments Analysis of DEGs

GO term and pathways enrichment analysis was performed using the R package clusterProfiler based on the GO and Kyoto encyclopedia of genes and genomes (KEGG) databases, respectively. The GO term or KEGG pathway with *p*-value less than 0.05 was significantly enriched.

### 2.7. Real-Time Quantitative PCR (RT-qPCR)

Total RNAs were extracted from J774A.1 cells with TRIzol reagent followed by cDNA synthesis with the reverse transcription kit (No.FSK-101, TOYOBO, Japan) containing Oligo (dT) primer (5′-TTTTTTTTTTTTTTTTTTTT-3′) according to the manufacturer’s protocol. Quantitative real-time PCR was performed using the SYBR Green Real-time PCR Master Mix (No.QPK-201, TOYOBO, Osaka, Japan) and a 7900HT Fast Real-Time PCR System (Applied Biosystems, Waltham, MA, USA). Amplification was performed at 50 °C for 2 min, 95 °C for 10 min, and then followed by 40 cycles of 95 °C for 15 s, 60 °C for 15 s, and 72 °C for 30 s. Primers used in the study were listed in Appendix A and synthesized by Sangon Biotech (Shanghai, China). The relative expression values of candidate mRNAs were normalized to that of GAPDH in each sample using the 2^−ΔΔCt^ method.

### 2.8. Immunofluorescence under Confocal Laser Scanning Microscopy

J774A.1 cells were seeded in glass bottom confocal chambers at approximately 1 × 10^6^ cells per chamber, cultured overnight until adhesion, then infected with PR8 virus and treated with SBE for 24 h. The cells were fixed with 4% paraformaldehyde (PFA) and permeabilized with 0.3% Triton X-100, washed three times with cold PBS buffer. Before the mouse polyclonal anti-NP (No. ab128193, Abcam, Cambridge, UK) or rabbit polyclonal anti-NF-kB p65 (No. D14E12, CST, Danvers, CO, USA) primary antibody was added, the cells were blocked with PBS supplemented with 5% goat serum (No. ZLI-9022, BIODEE, Beijing, China) for 1 h at room temperature. After incubation with the primary antibodies overnight at 4 °C, cells were then rinsed three times with PBS and followed by co-incubation of goat anti-mouse (No. ab150116, Abcam, Cambridge, UK) or goat anti-rabbit (No. ab150080, Abcam, Cambridge, UK) secondary antibody conjugated with Alexa Fluor 594 at room temperature in the dark for 1 h. After washing, DAPI solution (No. P0131, Beyotime, Shanghai, China) was added for nuclear staining at room temperature for 10 min, followed by washing three times using PBS buffer. The expression and localization of NP or NF-kB p65 were detected by immunofluorescence under FV3000 confocal laser scanning microscope (Olympus, Tokyo, Japan) with oil immersion under 100 × objective in 1 mL PBS buffer.

### 2.9. Western Blot Assay

J774A.1 cells were collected and lysed with RIPA lysate (No. R0010, Solarbio, Beijing, China) containing 1% PMSF, centrifuged at 12,000 rpm for 10 min at 4 °C. The supernatant was quantified by a BCA protein assay kit (No.cx00098, Beyotime, China). The proteins were subjected to SDS-PAGE electrophoresis after denaturation and transferred to PVDF membranes (No. ISEQ00010, Millipore, Billrica, MA, USA), then blocked in TBST solution with 5% non-fat milk (No. 9999S, CST, USA) for 1 h. The PVDF membranes were incubated with anti-NP (1:1000, mouse, No. ab128193, Abcam, England) and GAPDH (1:5000, mouse, No.60004-1, Proteintech, Rosemont, PA, USA) primary antibody diluent at 4 °C overnight, respectively. The membrane was washed with 10 mL TBST buffer (No. T1085, Solarbio, China) three times and followed by adding the secondary antibodies (No. SA00001-17, Proteintech, USA) that was conjugated with horseradish peroxidase (HRP) for 4 h at 4 °C. After another washing steps, the membrane was soaked with 0.2 mL ECL chemiluminescence substrate (No.34095, Thermo Fisher, USA) and visualized using ChemiDoc MP Imaging System (No.12003154, Bio-Rad, Hercules, CA, USA). The densitometry of the immunoblot was calculated using Image J software, and the ratio of NP to internal reference GAPDH was presented as a bar diagram.

### 2.10. Quantification and Statistical Analysis

Statistical analysis was performed with GraphPad Prism version 8.3 (GraphPad Software, Inc., San Diego, CA, USA). The immunofluorescence and immunoblot were analyzed using Image J software. The experimental results are shown as mean ± SD, and the student’s t test was used to compare the two groups. Differences were considered statistically significant if the *p*-value < 0.05. Significance levels are: * *p* < 0.05 and ** *p* < 0.01 vs. Mock group; ^#^
*p* < 0.05 and ^##^
*p* < 0.01 vs. PR8 group. Each set of experiments was independently repeated three times.

## 3. Results

### 3.1. SBE Plays Dual Role of Antivirus and Inflammatory Inhibition in Macrophage Infected with PR8 Strain

To determine the effective components of SBE used in this study, UPLC/Q-TOF MS technology was used to analyze the target compounds according to the rich structure information in the MS combined with the cleavage law of various chemical components. The results showed that the main components of SBE included Scutellarin, Baicalin, Chrysin-7-O-glucuronide, Oroxylin A-7-O glucuronide, Wogonoside, Salvigenin, and Wogonin, a total of 7 compounds were all flavonoids (Figure 1A and Appendix A).

To construct the cellular model of the inflammatory response caused by influenza A virus infection, J774A.1 macrophages were infected by the PR8 strain with 100 TCID_50_ (Figure 1C). The expression of the signature pro-inflammatory cytokines IL-1β and IL-6 was significantly increased, which revealed the virus induces an inflammatory response (Figure 1D,E). Moreover, the successful infection was confirmed by detecting the expression of viral NP through immunofluorescence, western blot, and RT-qPCR (Figure 2A–E).

We next examined the effect of SBE on J774A.1 cell infected with PR8 by treating the cells with the TC_0_ of 6.25 μg/mL of SBE (Figure 1B,C). It was observed that both the replication of viral NP (Figure 2A–E) and the production of cytokines IL-1β and IL-6 (Figure 1D,E) were effectively inhibited. These results indicate that the SBE solution could play dual pharmacological roles of inhibiting virus replication and controlling inflammatory reactions in macrophages infected with the PR8 strain.

### 3.2. Transcriptome Analysis of the Cellular Gene Expression after Virus Infection and Intervention Effect of SBE

To conduct a holistic study of the cellular response after PR8 virus infection and the effective mechanisms of the anti-inflammatory effect of SBE, transcriptome analysis experiments were performed to detect changes in global gene transcript levels (Figure 1C). Totally, ~178, ~130 and ~183 million raw sequencing reads were obtained from the mock, PR8 infected and SBE treatment with PR8 infected samples, respectively. After quality analysis of sequencing results using Fastp software by filtering out the adapter and low-quality sequences, we obtained clean reads ~170 million in the mock group, ~124 million in the PR8 group and ~174 million in the SBE group, respectively (Appendix A). All of the reads were mapped to the *Mus musculus* genome by HISAT2 software, and the total mapping ratios were 96.84%, 97.03% and 97.12% in the three groups, respectively (Appendix A), indicating the high quality of the sequencing data.

### 3.3. Differentially Expressed Genes (DEGs) Analysis

After quantifying the gene expression levels using fragments per kilobase of exon per million mapped reads (FPKM), principal component analysis (PCA) indicated that the samples in the same group were clustered closely together, indicating satisfactory reproducibility within the group and obvious specificity between the groups (Figure 3A and Appendix A).

Then the DEGs analysis was performed with DEseq2 software following the parameter of fold change > 2 and *p*-value < 0.05. Finally, compared to mock cells, 606 DEGs with 443 upregulated DEGs and 163 downregulated DEGs were identified in PR8 strain-infected cells (Figure 3B and Appendix A). Moreover, SBE intervention on PR8 infected cells resulted in 774 DEGs, of which 132 promoted DEGs and 642 suppressed DEGs (Figure 3B,C). Taken together, these data suggest that the infection of J774A.1 macrophage by PR8 strain induces extensive changes in the expression pattern of host genes, and the intervention of SBE also leads to widespread genetic alteration. A Venn diagram was generated to examine the unique and overlapping DEGs among the three groups. The number of overlapping DEGs was as high as 347, accounting for 33.6% of all DEGs (Appendix A). Interestingly, as many as 315 DEGs were induced by PR8 infection and decreased after SBE treatment, indicating that SBE could play a reverse transcriptional effect contrary to viral infection. After SBE treatment, a series of host genes were expressed in the downtrend compared to those induced by the PR8 virus. We suggest that the overlapping DEGs are involved in viral replication and PR8-induced inflammatory response and are also critical targets of SBE to exert viral inhibition and reverse inflammation.

### 3.4. SBE Inhibits the Activation of Innate Immune Response after Virus Infection Revealed by GO Enrichment Analysis

To further explore the biological functions of the DEGs after SBE treatment, GO enrichment analyses were performed to annotate the host genes by classifying molecular function, cellular component, and biological process. The genes enriched in the molecular function category are mainly associated with cytokine receptor binding, cytokine activity, receptor regulator activity, receptor-ligand activity, double-stranded RNA binding, and GTPase activity (Figure 4A,B). The genes enriched in cellular components were predicted to localize inside the membrane primarily, the external side of the plasma membrane, extracellular membrane-bounded organelle, symbiont containing vacuole (Figure 4C,D). Considering the molecular functions of DEGs, it is speculated that these SBE-intervened genes are involved in the pattern recognition, secretion of cytokines and acting on downstream cells, leading to the above localization analysis results. As for the biological process classification, the enrichment of DEGs was mainly related to the defense response to viruses, innate immune response regulation, cytokine production and cytokine-mediated signaling pathway, and response to interferon and adaptive immune response (Figure 4E,F). The results of GO analysis demonstrate that PR8 infection in the present experimental model induced an antiviral immune response, especially the production and effect of cytokines in the innate immune response, consistent with the report that IAV infection induced a strong “cytokine storm”. Meanwhile, SBE treatment can inhibit the production and effect of cytokine storms and regulate innate immunity to reverse excessive inflammatory response by affecting the expression of multiple genes.

### 3.5. SBE Interferes with Pattern Recognition Receptor Signaling Pathway Based on KEGG Pathway Enrichment Analysis

To identify the cellular pathways potentially involved during SBE treatment with PR8 infection, the pathway enrichment analysis of 315 DEGs was executed using the KEGG database. Combining the methods of ORA and GESA, in addition to the known virus-related responses, the significantly enriched pathways were mainly summarized into NOD-like receptor (NLR), cytosolic DNA-sensing, RIG-I-like receptor (RLR), Toll-like receptor (TLR), TNF, cytokine-cytokine receptor interaction and JAK-STAT signaling pathway (Figure 5A). Consistent with GO analysis, the KEGG pathways were enriched in innate immune-associated pathways, especially in pattern recognition receptors and their downstream pathways. The mechanism of SBE can be summarized as affecting the pattern recognition receptor (PRR) signaling pathway in macrophages, inhibiting a variety of cytokines production and receptor binding, intervening downstream signal transduction (JAK-STAT), which ultimately leads to a reduced inflammatory response by targeting multiple genes.

To further analyze the key targets of SBE action, we analyzed the significant enriched DEGs in the KEGG pathway according to the expression level and immune function (Figure 5B). In the present transcriptome profile of PR8 infection, TLR3 (~8.3 folds), TLR7 (~1.5 folds) and TLR8 (~8.6 folds) were stimulated, which locate on intracellular vesicles and specifically recognize dsRNA or ssRNA components from the viral genome or replication intermediate. On the contrary, SBE intervention significantly reduced the TLR3 (~13.9 folds), TLR7 (~4.2 folds) and TLR8 (~21.4 folds) transcription. Similarly, the RIG-I (also known as DDX58) and MDA5 (also known as IFIH1), as intracellular RNA recognition receptors in the RLR pathway, were induced respectively with 12.1 folds and 7.4 folds by PR8, while SBE inhibited their transcription with 17.0 folds and 7.9 folds, respectively. Among multiple receptors in the NLR pathway, SBE attenuated NLRP3 (~1.8 folds), NLRC4 (~3.6 folds) and NLRC5 (~3.4 folds) transcription. And as DNA sensors, cGAS (~2.0 folds), IFI202 (~3.1 folds), and IFI204 (~21.2 folds) mRNA levels were also inhibited by SBE. In addition to the receptor genes listed above, the transcriptional level of adaptor molecules MyD88 (~2.5 folds) in the TLR pathway, MAVS (~1.5 folds) in the RLR pathway, ASC (~2.5 folds) in NLR and AIM2 pathway, and the common transcription factor IRF7 (~38.8 folds) downstream of PRR were attenuated by SBE treatment. Finally, SBE effectively inhibited the effector molecules of the pattern recognition pathway, inflammatory cytokines IL-15 (~4.1 folds), IL-18 (~11.8 folds), and interferon-inducible gene IFI35 (~5.3 folds) and NMI (~4.4 folds) transcription (Figure 5B and Appendix A).

### 3.6. Validation of Key DEGs within PRR Signaling Pathway and Detection of Downstream Effect

To confirm the differential expression between the PR8-induced genes and SBE-reduced genes in sequencing data, a series of representative DEGs, including PRR (TLR3, TLR7, TLR8; RIG-I, MDA5; NLRP3, NLRC4, NLRC5; cGAS, IFI202 and IFI204), adaptors (MyD88, MAVS and ASC), transcript factor IRF7 and effect inflammatory factors (IL-10, IL-15, IL-18, IFI35 and NMI) were selected for RT-qPCR analyses (Figure 5C and Figure 6). The results in J774A.1 macrophage by RT-qPCR exhibited similar signatures as detected in the transcriptome data. This further validates the robustness of experimental setup and bioinformatic analyses and corroborates the regulation of the PRR signaling pathway and inflammatory response by PR8 induction and SBE treatment.

Furthermore, to verify the role of SBE in inhibiting the activation of PRR signaling pathways, NF-κB nuclear translocation was detected in three groups as a classical transcription marker downstream of the PRR. Immunofluorescence images displayed the NF-κB p65 subunit translocating severely into the nucleus induced by PR8 infection, while the proportion and degree of NF-κB p65 into the nucleus was decreased after SBE treatment (Figure 7A,B). Although the transcription level of NF-κB in transcriptome data did not differ, the localization and activation of NF-κB p65 in the nucleus significantly differed among the groups, indicating that the transcription level of its target genes also had the exact change.

The above experimental results prove that PR8 infection indeed overactivates TLR, RLR, NLR and DNA sensing pathways, induces transcription factor NF-κB nuclear translocations, and promotes the release of a variety of inflammatory factors, while SBE treatment can inhibit inflammatory response by acting on these pathways.

## 4. Discussion

### 4.1. The Over-Activation of PRR Pathways Leads to the Aggravation of Inflammation Induced by IAV Infection

PRRs serve as priming signals to initiate innate immune response via recognizing pathogenic associated molecular patterns (PAMPs) of foreign pathogens or damage-associated molecular patterns (DAMPs) generated under inflammatory conditions [23]. Influenza virus infection is recognized by a variety of PRRs [24,25]. In the early stage of viral invasion, PPRs bind corresponding ligands to activate signaling pathways for producing the interferons (IFNs), chemokines, pro-inflammatory cytokines, and other molecules to induce inflammatory responses, which is conducive to the clearance of viruses and infected cells. However, the excessive inflammation induced by continuous PRR pathways activation leads to inefficient virus clearance and irreversible organ damage in patients, thus aggravating its severity and mortality [26,27]. It has been reported that TLR3 knockout mice infected with influenza A virus (IAV) show higher viral titer and fewer inflammatory mediators, but wild-type (WT) mice still display higher mortality, which is related to TLR3 mediated signaling pathway appears to be more prone to induce inflammatory cytokines rather than IFNs [28]. Although TLR7 and TLR8 activate distinct pathways in monocytes, both can sense ssRNA from IAV and activate NF-κB in a MyD88-dependent manner, producing IFN and inflammatory cytokines [29]. In particular, TLR7 mainly induces the expression of cytokines that promote Th17 polarization, which effectively mediates tissue inflammation increase. It was found that RIG-I was activated as a primary PRR for IAV dsRNA and engaged to MVAS on mitochondria, initiating the expression of inflammatory cytokines and IFN-α/β, while MDA5 mainly functioned as a transcriptional inducer, which benefits the amplification of interferon-stimulated gene production [30,31]. NLRP3 in macrophages can directly sense IAV RNA or poly (I:C), induce pyroptosis and secrete several mature IL-1β and IL-18. The expression levels of NLRP3, caspase-1, pro-IL-1β and pro-IL-18 were also upregulated after IAV infection depending on the NF-κB signaling pathway, which resulted in the formation of the inflammasome in the caspase-1-dependent pathway [32]. Although IAV components do not directly activate DNA sensor cGAS, the mtDNA released by damaged mitochondria and the micronuclei released by damaged nuclei induced by IAV infection are the main drivers of pathological type I IFN response [33].

In this study, transcriptome sequencing and analysis of macrophages were performed, and it found that transcription levels of the gene in the PRR signaling pathway, including NLR, RLR, TLR and DNA sensor signaling pathways, were extensively induced by IAV infection (Figure 5A). The molecular functions, cellular components, and biological processes of DEGs analyzed by GO enrichment analysis were also consistent with the pathway obtained by KEGG (Figure 4). RT-qPCR also confirmed that the receptors, adaptors, and effector molecules on the above PRR signaling pathways were significantly induced for transcription (Figure 6). Hence, effective regulation and balance of the PRR signaling pathway may become a key strategy for influenza treatment. Several therapies, including small molecules based on targeting TLRs, RLRs and NLRs, are already under development [34,35,36,37,38]. However, these drugs all act on a single molecule target, and their toxic and side effects are unclear. Therefore, multi-target, safe and effective drugs should be a topic more widely screened and studied in the next few years.

### 4.2. SBE Effectively Reverses the Over-Transcription of Genes in PRR Signaling Pathways and Inflammatory Response Induced by IAV through Multiple Targets

*Scutellaria baicalensis* has been clinically used in disease treatment such as acute respiratory infection, acute gastroenteritis, infantile diarrhoea, trachoma hepatitis, hypertension, vomiting during pregnancy and other diseases [9,10]. Flavonoids and their glycosides are the main active components in *Scutellaria baicalensis*, and more than 40 kinds of flavonoids have been identified, among which baicalin, baicalein, wogonoside and wogonin have high contents and obvious pharmacological effects [39]. In the SBE used in this paper, seven major compounds, including Scutellarin, Baicalin, Chrysin-7-O-glucuronide, Oroxylin A-7-O glucuronide, Wogonoside, Salvigenin and Wogonin were identified (Figure 1A). In treating respiratory tract injury caused by IAV infection, studies have found that *Scutellaria baicalensis* and its active components can inhibit the cell damage directly caused by virus replication and reduce the host’s immune and inflammatory response [14,15].

This study confirmed that SBE could inhibit the replication of influenza virus in the macrophage (Figure 2). Previous studies suggested *Scutellaria baicalensis* and its components played the antiviral role by blocking neuraminidase (NA) activity, inhibiting autophagy and induction of IFN [18,40,41]. In addition, transcriptome data and experiments in this study were used to confirm that SBE inhibits influenza virus-induced inflammation through multiple targets and pathways, especially the PRR signaling pathway. Firstly, we screened out that the main targets of SBE were involved in the TLR, RLR, NLR and DNA sensor signaling pathway through the combined GO and KEGG pathway enrichment analysis of 315 DEGs reversed by SBE (Figure 4 and Figure 5A). Then, SBE was determined to significantly inhibit the mRNA transcription of PRRs (TLR3, TLR7, TLR8, RIG-I, MDA5, NLRP3, NLRP4, NLRC5, ZBP-1, cGAS, IFI202, IFI204), adaptors (MyD88, MAVS, ASC) and transcript factor IRF7 and pro-inflammatory molecules (IL-15, IL-18, IL-27, IFI35 and NMI) based on the RNA-sequence data combined with RT-qPCR validation (Figure 5B,C, Figure 6 and Appendix A). Finally, the inhibitory effect of SBE on PRR signaling pathway activation and inflammatory response was verified by the levels of IL-1, IL-6 and the translocation into the nucleus of NF-κB p65 (Figure 1D,E and Figure 7). Previous studies have reported that *Scutellaria baicalensis* and its active components inhibit immune inflammatory responses and reduce pathological damage by regulating the synthesis and release of pro-inflammatory cytokines, inhibiting inflammatory mediators, anti-oxidant and scavenging free radicals [9,10]. Our study suggests that the anti-inflammatory effect of SBE is mainly related to the inhibition of PRR activation and effector molecules production, which further elucidates the pharmacologic mechanism of *Scutellaria baicalensis* at the molecular level and provides therapeutic targets, which could be used to reverse the inflammation caused by IAV.

### 4.3. Multi-Target Reversal Drug Screening and Development Based on Gene Transcription Map of IAV Infection

The outcome of viral infectious diseases depends on the interaction between the host and the virus, while the biggest harm is the tissue damage caused by the excessive immune and inflammatory pathological response. Therefore, treating IAV infection should actively resist the virus in the early stage of infection and prevent the excessive inflammatory response [8]. *Mosla scabra* flavonoids [42] and *Alstonia scholaris* alkaloids [43] were found to significantly inhibit IAV replication and attenuate IAV-induced lung inflammation by regulating the PRR signaling pathway. High-throughput omics technology can be used to widely screen more natural product-derived drugs for the prevention and treatment of IAV infection. RNA sequencing technologies provide an understanding of viral infection and host interactions, especially the global and systematic changes in transcriptional maps. Using a gene transcription map as a bridge to connect the diseases and drugs, then drugs can be screened or found to reverse the map on the pathogenesis of diseases. Most studies on RNA sequencing have analyzed the role of non-coding RNA in the pathogenesis of IAV [44,45,46]. Studies have also found that the influenza infection induced in the nasal epithelium early and altered responses in interferon gamma signaling, B-cell signaling, apoptosis, necrosis, smooth muscle proliferation, and metabolic alterations [47,48,49]. Through transcriptome sequencing in a macrophage model, we found that SBE could reverse IAV-induced transcriptional map changes, especially the key genes in PRR signaling pathways (Figure 3 and Appendix A). Traditional Chinese medicine of natural origin is often characterized by multi-component, multi-target, multi-pathway, and low toxicity to prevent and treat diseases, which is an advantageous source for drug screening of reverse transcription map. The transcriptome data of the viral infection model and drug intervention could be analyzed to develop antiviral drugs or study the mechanism.

### 4.4. Intervention Macrophage and “Cytokine Storm” Is Crucial for Treatment of Viral Pneumonia

Macrophages are the key cells that resist influenza infection and initiate the inflammatory response. Resident alveolar macrophages not only play the role of phagocytosis of pathogens but also recruit neutrophils and other immune cells to the lesion site, which may lead to excessive inflammatory response and damage while defending against viruses. Although pulmonary interstitial macrophages had no significant effect on viral titers, they increased pro-inflammatory cytokines and pulmonary monocytes [50]. Alveolar macrophages derived from peripheral blood monocytes can persist in the acute inflammatory environment caused by IAV and increase the secretion of various inflammatory cytokines [51]. Accumulation and activation of macrophages in lung tissue led to an excessive immune response and even a deadly “cytokine storm”. IAV infection can lead to a “cytokine storm”, and the release level of various cytokines is closely related to the degree of disease [52,53]. In this study, we found that SBE could inhibit the transcription of IL-1, IL-6, IL-10, IL-15, and IL-18 in macrophages (Figure 6 and Appendix A), among which IL-1, IL-6, IL-15 and IL-18 are typical pro-inflammatory cytokines, while IL-10 is considered as an anti-inflammatory cytokine [54,55]. IL-10, downstream of IL-6, is also one of the markers of the “cytokine storm” induced by IAV infection, indicating that the immune system can maintain the balance of inflammatory response through the interaction between different cytokines [56]. In this paper, reducing IL-10 also reflects the effect of SBE in alleviating the excessive release of various cytokines.

Furthermore, we also found that SBE significantly relieves the transcription of interferon-inducible gene NMI and IFI35 (Figure 6 and Appendix A). Studies have confirmed that NMI and IFI35 released by macrophages can aggravate the inflammatory response as DAMP, and the gene knockout mice have significantly reduced inflammation and improved survival rate in sepsis [57]. Moreover, neutralizing antibodies against IFI35 greatly ameliorated lung injury and mortality in IAV-infected mice [58].

In humans, most of the studies employed in vitro culture of alveolar and blood monocyte-derived macrophages. Influenza virus replication and induction of proinflammatory cytokine responses were much stronger in monocyte-derived macrophages [59]. By comparing a variety of human cells, it was found that the monocytic cell U937 supports the replication of influenza viruses as well as the production of critical pro-inflammatory cytokines, such as IL-6, IL-8, TNFα, IFN, MCP-1, IP-10, and MIP-1 [60,61]. These cytokines have been detected in patients with severe influenza [62]. This study focused on murine macrophages, but given the genetic similarities between mice and humans, especially the conserved functions of TLR3/7/8, RIG-I/MDA5, NLRP3 and cGAS signaling pathways in mice and humans [59], the pharmacological effects of SBE found in this study may also apply to human cells. There are few studies on the effect of SBE on influenza virus-infected human macrophages. In the LPS-induced THP-1 macrophage model, *Scutellaria baicalensis* pith-decayed root reduced caspase-1 activation and IL-1 expression by inhibiting the NF-κB/NLRP3 pathway [63]. We propose that further studies in humans are needed to decipher the role of SBE against influenza virus-induced innate immunity, particularly in macrophages. But one thing is certain the regulation of macrophages and cytokines can be used as a therapeutic strategy for IAV infectious diseases to reduce cell death and destruction in normal lung tissues.

## 5. Conclusions

In conclusion, we determined that SBE with seven flavonoids as the main component could play dual roles of anti-influenza virus and anti-inflammatory by using a macrophage model in vitro. Combined with transcriptome sequencing analysis and experimental verification, this study reveals that the anti-inflammatory effect of SBE was related to its inhibition of PRR pathway activation. These findings provide therapeutic targets for influenza virus infectious diseases and an experimental basis for the clinical application of SBE or its main components. Of course, further studies are needed to identify key targets of SBE and to conduct more comprehensive and systematic validation in animal models in vivo.

## Figures and Tables

**Figure 1 viruses-15-01524-f001:**
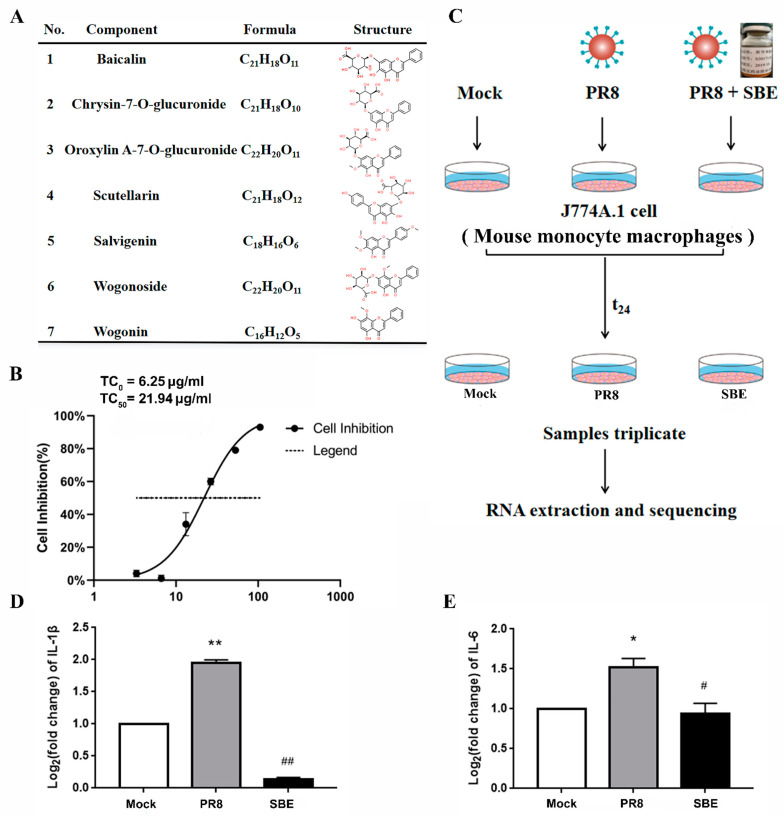
Establishment and determination of *Scutellaria Scutellaria* extract (SBE) interference with inflammation in cultured J774A.1 macrophages induced by influenza A virus. (**A**) The compounds, chemical formulas, and structures of the main components of SBE used in these experiments determined by UPLC/Q-TOF MS are listed. (**B**) The maximum nontoxic concentration (TC_0_) and half-maximal toxic concentration (TC_50_) of SBE on J774A.1 cell was determined according to the cell inhibition rate detected by CCK8. (**C**) Workflow of PR8 infection, SBE intervention and transcriptome analysis in mice J774A.1 cell line. (**D**) The mRNA levels of representative pro-inflammatory cytokine interleukin (IL)-1β in Mock, PR8 and SBE groups were detected by RT-qPCR. (**E**) The mRNA levels of pro-inflammatory cytokine IL-6 in Mock, PR8 and SBE groups were also detected by RT-qPCR. GAPDH was used as an internal control. * *p* < 0.05, ** *p* < 0.01 vs. Mock group; ^#^
*p* < 0.05, ^##^
*p* < 0.01 vs. PR8 group.

**Figure 2 viruses-15-01524-f002:**
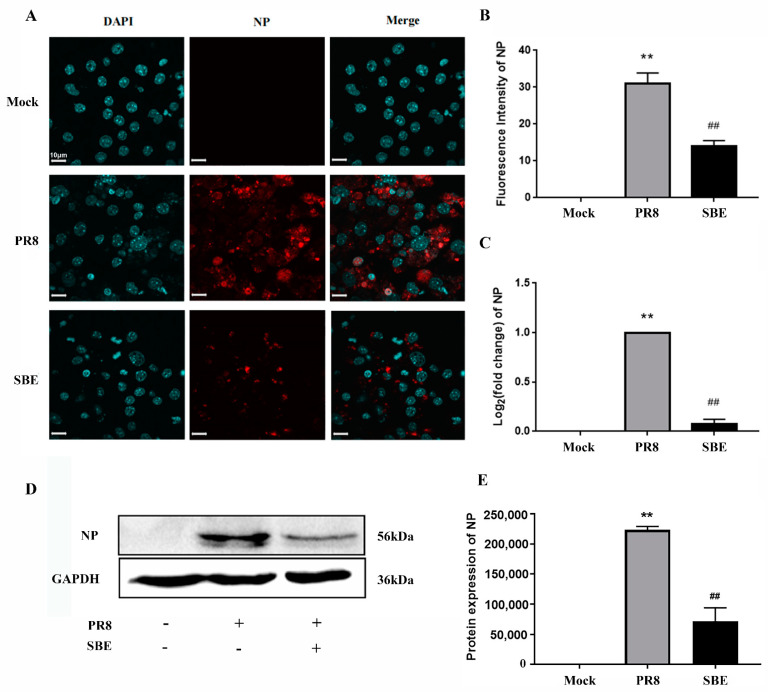
Detection of influenza A virus replication and identification of inhibition after treatment with SBE in J774A.1 macrophage. (**A**) Representative images of immunofluorescence staining for the viral nucleoprotein (NP) in Mock, PR8 and SBE groups were observed by confocal laser scanning microscope. The nuclei are shown in blue using DAPI staining, while the NP are shown in red and detected by the anti-NP antibody. (**B**) Bar diagram of the quantitative summary for NP immunofluorescence. Around 100 cells in each group were counted. (**C**) The mRNA levels of NP in Mock, PR8 and SBE groups were detected by RT-qPCR. GAPDH was used as an internal control. (**D**) The expression levels of NP in Mock, PR8 and SBE groups were detected by western blot using GAPDH as an internal control. (**E**) The statistical bar graph of the relative densitometry of immunoblotting bands of NP. ** *p* < 0.01 vs. Mock group; ^##^ *p* < 0.01 vs. PR8 group.

**Figure 3 viruses-15-01524-f003:**
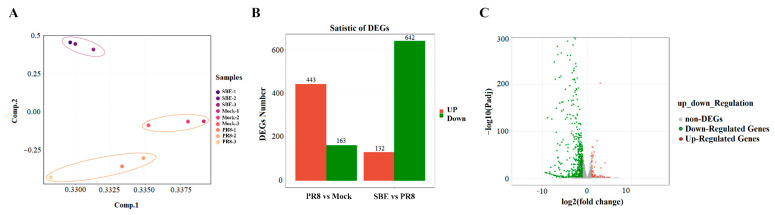
Sample analysis and differentially expressed genes (DEGs) analysis of PR8-infected and SBE intervention macrophages. (**A**) principal component analysis (PCA) analysis. The principal component of each sample was analyzed considering the gene expression in the corresponding samples. Samples corresponding to each experimental group (three biological replicates per group) were plotted on the first two principal components. (**B**) Numbers of DEGs in PR8 group to Mock group and SBE group to PR8 group. Red represents up-regulated genes, and green represents down-regulated genes. The *Y*-axis indicates the number of genes. (**C**) Volcano plots of the distribution of DEGs between SBE and PR8 groups. The red, green, and gray dots represent up-regulated, down-regulated, and not significantly regulated genes. The *X*-axis indicates the Log_2_ (fold change) of DEGs. The *Y*-axis represents the statistical significance, −Log_10_ (*p*-value).

**Figure 4 viruses-15-01524-f004:**
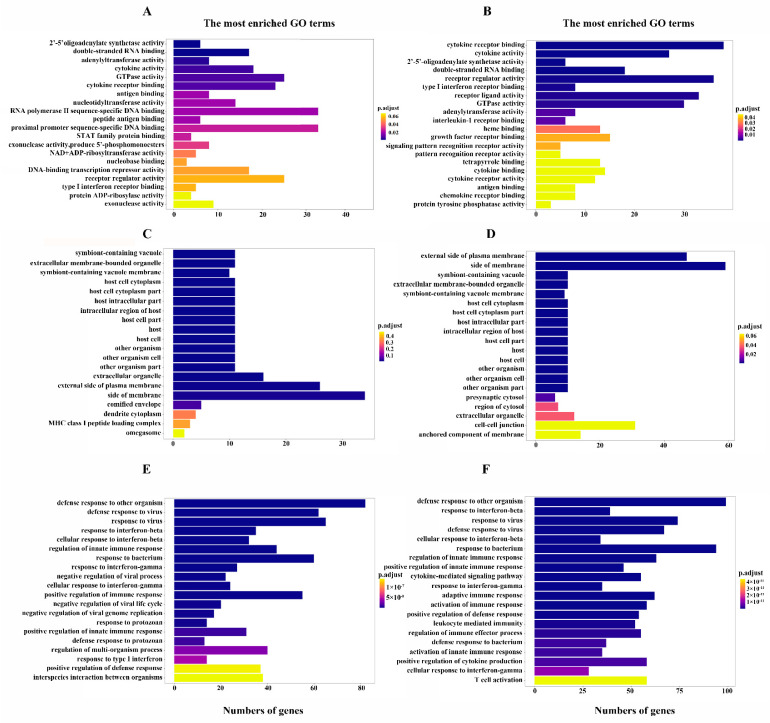
The bar chart of Gene Ontology (GO) enrichment analysis is based on the identified DEGs in J774A.1 macrophage in response to PR8 infection and SBE intervention. The top 20 significant GO terms categorized into a molecular function (**A**), cellular component (**C**) and biological process (**E**) were obtained by comparison between the PR8 group and Mock group. The top 20 significant GO terms categorized into a molecular function (**B**), cellular component (**D**) and biological process (**F**) were identified by comparison between the SBE group and PR8 group. The Y-axis indicates GO terms, the X-axis represents the counts of DEGs mapping to the corresponding GO terms, and the color corresponds to statistical significance.

**Figure 5 viruses-15-01524-f005:**
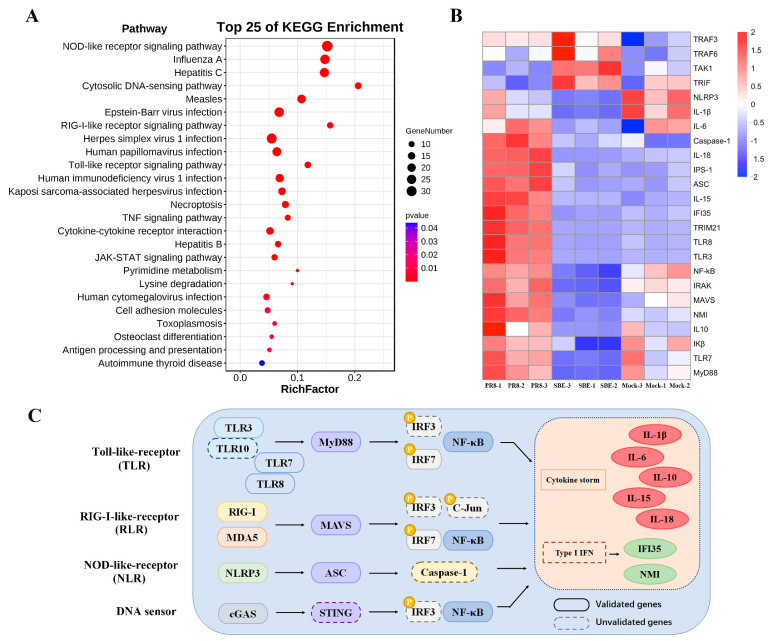
Analysis of key pathways and genes of SBE intervention in J774A.1 macrophage model infected with PR8 virus. (**A**) A dot map showing the compilation of the top 25 enriched Kyoto encyclopedia of genes and genomes (KEGG) pathways was determined from 315 DEGs of intersection between Mock, PR8 and SBE groups. The transcriptional level of 315 DEGs was increased in the PR8 group compared to the Mock group and decreased in the SBE group compared to the PR8 group. The *X*-axis indicates the gene ratio corresponding to the pathway; the Y-axis indicates pathway terms; the circle size reflects the number of gene enrichment; and the color corresponds to statistical significance. (**B**) Heatmaps of key DEGs related to pattern recognition receptor pathways upon PR8 infection and SBE intervention were visualized. The color shows the fold change of detected genes. (**C**) Schematic representation of the signaling pathways and key genes that converge on Toll-like receptor (TLR), RIG-I-like receptor (RLR), NOD-like receptor (NLR), and DNA sensor. The genes detected and validated are shown in the solid boxes, whereas the genes that remained undetected or unvalidated in this study are shown in the dotted boxes.

**Figure 6 viruses-15-01524-f006:**
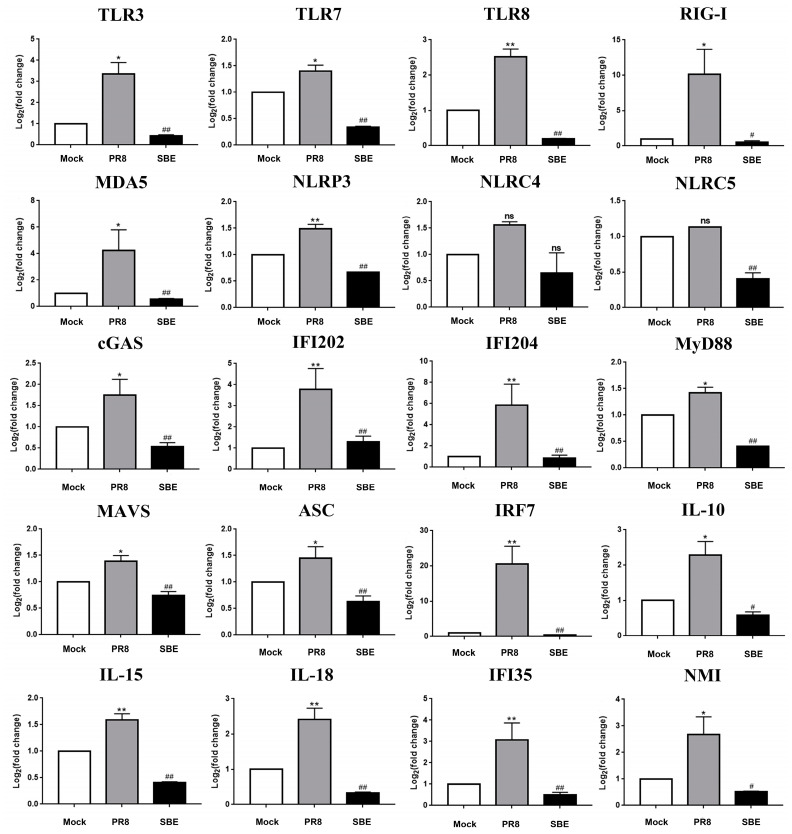
Validation of representative DEGs involved in pattern recognition pathways by RT-qPCR. The mRNA levels of pattern recognition receptors (TLR3, TLR7, TLR8, RIG-I, MDA5, NLRP3, NLRC4, NLRC5, cGAS, IFI202 and IFI204), adaptors (MyD88, MAVS, ASC), transcript factor IRF7 and downstream effect genes (IL-10, IL-15, IL-18, IFI35 and NMI) in Mock, PR8 and SBE groups were detected by RT-qPCR. GAPDH was used as an internal control. Data were obtained from three biological replicates. * *p* < 0.05, ** *p* < 0.01 vs. Mock group; ^#^
*p* < 0.05, ^##^
*p* < 0.01 vs. PR8 group; ns indicates no statistical significance.

**Figure 7 viruses-15-01524-f007:**
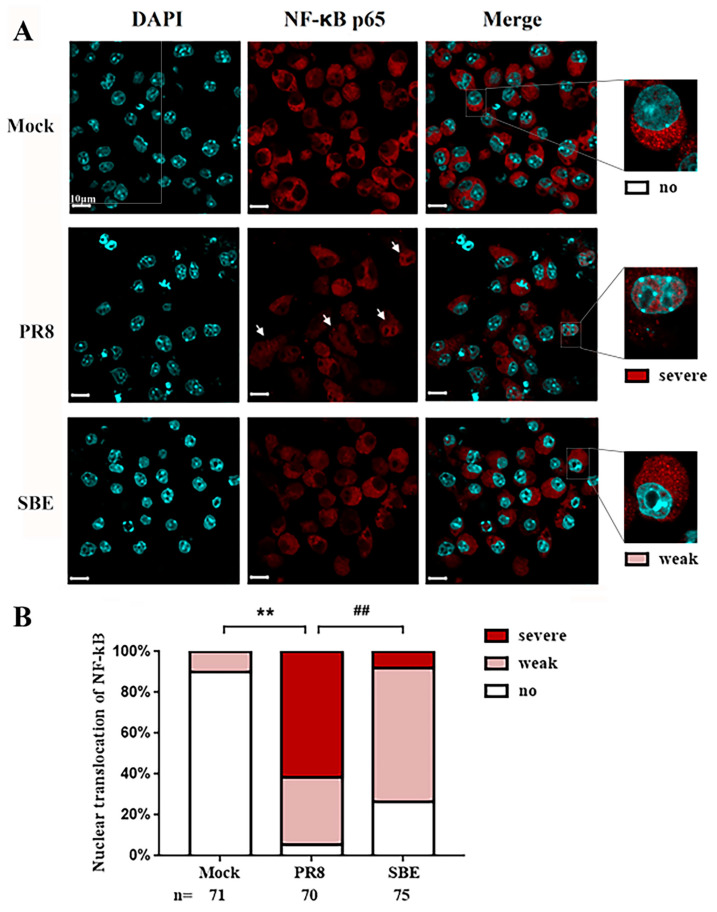
Detection of nuclear translocation of NF-κB p65 after treatment with SBE in J774A.1 macrophage infected by PR8 virus. (**A**) Representative images of immunofluorescence staining for the NF-κB p65 in Mock, PR8 and SBE groups were observed by confocal laser scanning microscopy. The nuclei are shown in blue using DAPI staining, while the NF-κB p65 are shown in red and detected by the antibody. The white arrows indicate cells where the NF-κB p65 is clearly transported to the nucleus. (**B**) The ratio of cells with nuclear translocation of NF-κB p65 subunit. The red, pink, and white bars represent severe, weak, and no NF-κB p65 nuclear translocation. The number of cells counted for each group is listed below. ** *p* < 0.01 vs. Mock group; ^##^ *p* < 0.01 vs. PR8 group.

## Data Availability

All data generated or analyzed in the current study are included in this published article and its Appendix A. Please contact the corresponding author for details.

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
