# Peer review of "The Extract of Scutellaria baicalensis Attenuates the Pattern Recognition Receptor Pathway Activation Induced by Influenza A Virus in Macrophages"

_viruses, 2023, doi:10.3390/v15071524_

Round 1

Reviewer 1 Report

The manuscript by Yang et al is a comprehensive and informative report on the effects of a Scutellaria baicalinase extract on gene expression in murine macrophages infected with influenza A virus. Although SBE has been previously reported to have both antiviral and anti-inflammatory effects, the present study approaches the molecular aspects in a logical and comprehensive manner identifying changes at molecular level using cutting edge technologies. 

The manuscript is well written, the experiments and results are clearly delineated, and the conclusions are sound. The only issue is the limitation to the murine macrophage J774 cell line. The impact of the study would have been significantly increased by having some results with human macrophages, especially in terms of relevance for the development of SBE treatment for IAV caused pneumonia. This issue should be addressed at least in Discussion.

Reviewer 2 Report

The authors of presented study analysed the antiviral and anti-inflammatory effects of the extract from Scutellaria Baicalinase (SBE) in macrophages infected with influenza a virus. The performed the transcriptome analyses using high-throughput RNA sequencing. They identified 315 upregulated genes after infection with influenza virus. Treatment with BSE resulted in stabilization of transcription of these genes. SBE intervened genes are involved in the pattern recognition, secretion of cytokines and cytokines regulated pathway, response to interferons and adaptive immune response. The results are presented clearly, the manuscript is well written. The manuscript provides original data and provides an essential experimental basic for the application of SBE and its components in the clinical treatment of viral pneumonia.  I have some minor comment:

-          Figure 1A is too small. It is not readable

-          Figure 2B, 2D, 2E low quality of the pictures. Legends are not readable.

-          Figure 4 – pictures are too small and thereby it is not possible to read the description in the pictures – this information is important.

-          Figure 6 – low quality of the graphs. The legends in the pictures are too small and are not readable.  

Reviewer 3 Report

Recently, the development of therapeutic drugs for influenza and other respiratory viruses has increasingly focused on the dual strategy of inhibiting the viral replication cycle and reducing the inflammatory response.

This manuscript by Mingrui Yang and co-workers describes the ability of a Scutellaria baicalinase extract to inhibit viral replication and inflammatory response in influenza virus-infected macrophages.

The authors identified 315 genes whose transcript levels increase after infection and are reduced by treatment with Scutellaria baicalinase extract. In particular Scutellaria baicalinase extract inhibits the nuclear translocation of NF-κB p65 by reversing the virus-induced activation of the pattern recognition receptor signaling pathway and inflammation in macrophages.

The authors state that their study “provides a basic experimental basis for the application of Scutellaria baicalensis and its major components in the clinical treatment of viral pneumonia”.

This statement seems to be misplaced because there are already numerous studies that have described this application. Furthermore, the active fractions of the extract, which are only listed in the present study, have already been identified and thoroughly studied.

General Comments:

The study is not particularly innovative and the article has some flaws.

It is often difficult to follow the text because the article is not written correctly and this leads to difficulties in scientific evaluation.

It is not clear why the authors determine the principal components of Scutellaria baicalinase extract (7?) but then fail to study their role in Scutellaria baicalinase activity. Therefore, Figure 1A and Table S2 appear to be useless.

More than 80 compounds are known to have been isolated from Scutellaria baicalensis, including flavonoids (free and glycosylated forms) and phenylethanoid glycosides. For example, in previous research (Ji et al., 2015) 30 compounds of Scutellaria baicalensis were isolated and identified and, between these, several free flavones, including baicalein, wogonin and oroxylin A, showed potent anti-H1N1 activity. In particular: “Baicalin (15), baicalein (26), wogonin (27), chrysin (28) and oroxylin A (30) showed IC50 values of 7.4, 7.5, 2.1, 7.7 and 12.8 μM, respectively, which were remarkably more potent than the positive drug Osv-P (oseltamivir phosphate or Tamiflu, IC50 45.6 μM)”.

Mingrui Yang and co-workers instead describe the activity exclusively of the Scutellaria baicalinase extract. This activity, in particular the activity of baicalin, has already been known on both influenza virus (Nayak et al., 2014, Ding et al., 2014; Geng et al., 2020; Ma et al., 2021) and other viruses (Shi et al., 2016; Li et al, 2000; Liu et al., 2021).

Ding Y, Dou J, Teng Z, Yu J, Wang T, Lu N, Wang H, Zhou C. Antiviral activity of baicalin against influenza A (H1N1/H3N2) virus in cell culture and in mice and its inhibition of neuraminidase. Arch Virol. 2014 Dec;159(12):3269-78. doi: 10.1007/s00705-014-2192-2. Epub 2014 Jul 31. PMID: 25078390.

Geng P, Zhu H, Zhou W, Su C, Chen M, Huang C, Xia C, Huang H, Cao Y, Shi X. Baicalin Inhibits Influenza A Virus Infection viaPromotion of M1 Macrophage Polarization. Front Pharmacol. 2020 Oct 6;11:01298. doi: 10.3389/fphar.2020.01298. PMID: 33117149; PMCID: PMC7574031.

Ji S, Li R, Wang Q, Miao WJ, Li ZW, Si LL, Qiao X, Yu SW, Zhou DM, Ye M. Anti-H1N1 virus, cytotoxic and Nrf2 activation activities of chemical constituents from Scutellaria baicalensis. J Ethnopharmacol. 2015 Dec 24;176:475-84. doi: 10.1016/j.jep.2015.11.018. Epub 2015 Nov 11. PMID: 26578185.

Li BQ, Fu T, Dongyan Y, Mikovits JA, Ruscetti FW, Wang JM. Flavonoid baicalin inhibits HIV-1 infection at the level of viral entry. Biochem Biophys Res Commun. 2000 Sep 24;276(2):534-8. doi: 10.1006/bbrc.2000.3485. PMID: 11027509.

Liu H, Ye F, Sun Q, Liang H, Li C, Li S, Lu R, Huang B, Tan W, Lai L. Scutellaria baicalensis extract and baicalein inhibit replication of SARS-CoV-2 and its 3C-like protease in vitro. J Enzyme Inhib Med Chem. 2021 Dec;36(1):497-503. doi: 10.1080/14756366.2021.1873977. PMID: 33491508; PMCID: PMC7850424.

Ma QH, Ren MY, Luo JB. San Wu Huangqin decoction regulates inflammation and immune dysfunction induced by influenza virus by regulating the NF-κB signaling pathway in H1N1-infected mice. J Ethnopharmacol. 2021 Jan 10;264:112800. doi: 10.1016/j.jep.2020.112800. Epub 2020 Mar 27. PMID: 32224195.

Nayak MK, Agrawal AS, Bose S, Naskar S, Bhowmick R, Chakrabarti S, Sarkar S, Chawla-Sarkar M. Antiviral activity of baicalin against influenza virus H1N1-pdm09 is due to modulation of NS1-mediated cellular innate immune responses. J Antimicrob Chemother. 2014 May;69(5):1298-310. doi: 10.1093/jac/dkt534. Epub 2014 Jan 23. PMID: 24458510.

Shi H, Ren K, Lv B, Zhang W, Zhao Y, Tan RX, Li E. Baicalin from Scutellaria baicalensis blocks respiratory syncytial virus (RSV) infection and reduces inflammatory cell infiltration and lung injury in mice. Sci Rep. 2016 Oct 21;6:35851. doi: 10.1038/srep35851. PMID: 27767097; PMCID: PMC5073294.

The main bioactive compounds of Scutellaria baicalensis capable of counteracting the production of inflammatory cytokines were also thoroughly studied (see review Liao et al., 2020).

Moreover, the activity of Scutellaria baicalensis flavonoids for the treatment of pathologies in which the cellular immune response must be suppressed has already been described (Syafni et al., 2021).

Finally, the therapeutic capacity of baicalin in a variety of pulmonary diseases, including those of infectious origin, is widely known (Song et al., 2023; Wang et al., 2023).

Liao H, Ye J, Gao L, Liu Y. The main bioactive compounds of Scutellaria baicalensis Georgi. for alleviation of inflammatory cytokines: A comprehensive review. Biomed Pharmacother. 2021 Jan;133:110917. doi: 10.1016/j.biopha.2020.110917. Epub 2020 Nov 17. PMID: 33217688.

Song S, Ding L, Liu G, Chen T, Zhao M, Li X, Li M, Qi H, Chen J, Wang Z, Wang Y, Ma J, Wang Q, Li X, Wang Z. The protective effects of baicalin for respiratory diseases: an update and future perspectives. Front Pharmacol. 2023 Mar 16;14:1129817. doi: 10.3389/fphar.2023.1129817. PMID: 37007037; PMCID: PMC10060540.

Syafni N, Devi S, Zimmermann-Klemd AM, Reinhardt JK, Danton O, Gründemann C, Hamburger M. Immunosuppressant flavonoids from Scutellaria baicalensis. Biomed Pharmacother. 2021 Dec;144:112326. doi: 10.1016/j.biopha.2021.112326. Epub 2021 Oct 13. PMID: 34653757.

Wang D, Li Y. Pharmacological effects of baicalin in lung diseases. Front Pharmacol. 2023 Apr 24;14:1188202. doi: 10.3389/fphar.2023.1188202. PMID: 37168996; PMCID: PMC10164968.

Based on these observations, the manuscript does not seem to add much to what is already known.

Specific comments:

I have the following suggestions and comments.

Line 211: 

Please substitute “macrophage was infected by PR8 strain with …..” with: macrophages were infected with PR8 … 

Line 222: 

Figure 1A: Why list SBE members when they are not being studied as individual agents? Figure 1C is difficult to understand. What does cell inhibition mean? The non-cytotoxic or 50% cytotoxic concentrations were determined. Please swap Figure 1B with Figure 1C.

References:

Regarding influenza-induced inflammation and antiviral therapy some references such as:

Malik G, Zhou Y. Innate Immune Sensing of Influenza A Virus. Viruses. 2020 Jul 14;12(7):755. doi: 10.3390/v12070755. 

Wong JP, Christopher ME, Viswanathan S, Karpoff N, Dai X, Das D, Sun LQ, Wang M, Salazar AM. Activation of toll-like receptor signaling pathway for protection against influenza virus infection. Vaccine. 2009 May 26;27(25-26):3481-3. doi: 10.1016/j.vaccine.2009.01.048.  

Wong JP, Christopher ME, Viswanathan S, Dai X, Salazar AM, Sun LQ, Wang M. Antiviral role of toll-like receptor-3 agonists against seasonal and avian influenza viruses. Curr Pharm Des. 2009;15(11):1269-74. doi: 10.2174/138161209787846775. 

Dai X, Zhang L, Hong T. Host cellular signaling induced by influenza virus. Sci China Life Sci. 2011 Jan;54(1):68-74. doi: 10.1007/s11427-010-4116-z. 

Yu CH, Yu WY, Fang J, Zhang HH, Ma Y, Yu B, Wu F, Wu XN. Mosla scabra flavonoids ameliorate the influenza A virus-induced lung injury and water transport abnormality via the inhibition of PRR and AQP signaling pathways in mice. J Ethnopharmacol. 2016 Feb 17;179:146-55. doi: 10.1016/j.jep.2015.12.034. 

Green RR, Wilkins C, Pattabhi S, Dong R, Loo Y, Gale M Jr. Transcriptional analysis of antiviral small molecule therapeutics as agonists of the RLR pathway. Genom Data. 2016 Feb 1;7:290-2. doi: 10.1016/j.gdata.2016.01.020. 

Pattabhi S, Wilkins CR, Dong R, Knoll ML, Posakony J, Kaiser S, Mire CE, Wang ML, Ireton RC, Geisbert TW, Bedard KM, Iadonato SP, Loo YM, Gale M Jr. Targeting Innate Immunity for Antiviral Therapy through Small Molecule Agonists of the RLR Pathway. J Virol. 2015 Dec 16;90(5):2372-87. doi: 10.1128/JVI.02202-15. 

Zhou HX, Li RF, Wang YF, Shen LH, Cai LH, Weng YC, Zhang HR, Chen XX, Wu X, Chen RF, Jiang HM, Wang C, Yang M, Lu J, Luo XD, Jiang Z, Yang ZF. Total alkaloids from Alstonia scholaris inhibit influenza a virus replication and lung immunopathology by regulating the innate immune response. Phytomedicine. 2020 Oct;77:153272. doi: 10.1016/j.phymed.2020.153272.

should be added.

Supplementary Materials: 

Please swap Table S1 with Table S2

Language

The language of the manuscript must be improved. 

The language needs improvement. 

Round 2

Reviewer 3 Report

The authors responded satisfactorily to the referee's requests.

English has been improved